

# Comments on a state-operator correspondence for the torus

**Alexandre Belin, Jan de Boer and Jorrit Kruthoff**

Institute for Theoretical Physics, University of Amsterdam, Science Park 904,
Postbus 94485, 1090 GL Amsterdam, The Netherlands

## Abstract

We investigate the existence of a state-operator correspondence on the torus. This correspondence would relate states of the CFT Hilbert space living on a spatial torus to the path integral over compact Euclidean manifolds with operator insertions. Unlike the states on the sphere that are associated to local operators, we argue that those on the torus would more naturally be associated to line operators. We find evidence that such a correspondence cannot exist and in particular, we argue that no compact Euclidean path integral can produce the vacuum on the torus. Our arguments come solely from field theory and formulate a CFT version of the Horowitz-Myers conjecture for the AdS soliton.



# 1 Introduction

What data is needed to fully specify a conformal field theory? This remains one of the most important questions in conformal field theory, or more generally in quantum field theory. First, this data cannot be arbitrary as field theories are constrained by first principles. For example, unitarity, causality and locality give stringent constraints. The common lore is that the data needed to specify a CFT is the spectrum of local operators, namely the list of all operators and their respective conformal dimension, as well as the OPE coefficients. One can then compute an arbitrary correlation function of local operators by use of the OPE. To be more precise, one can compute an arbitrary correlation function on the Euclidean plane $\mathbf{R}^d$.

The conformal dimensions and OPE coefficients cannot be picked arbitrarily. Unitarity and crossing symmetry strongly constrain these parameters. By solving the solutions to the crossing equations, one can exclude large regions of parameter space. This program is known as the conformal bootstrap [1–4] (see [5] for a more recent review). If some data satisfies the crossing equations, one says that the set of correlation functions is consistent. But is this enough to fully specify a conformal field theory? A natural question to ask, is whether this is enough to fully specify the theory on an arbitrary manifold $\mathcal{M}$ not conformally related to $\mathbf{R}^d$.

For two-dimensional CFTs this question has been solved for rational conformal field theories [6, 7]. Moore and Seiberg showed that crossing symmetry was not sufficient, but that it had to be supplemented with modular covariance of the one-point functions on the torus. If crossing symmetry on the plane and modular covariance on the torus are satisfied, then we can put the theory on an arbitrary Riemann surface, on which it will be both well-defined and modular invariant. Unfortunately, an equivalent statement is lacking in higher dimensions. The reason this procedure is successful in two dimensions is that any Riemann surface can be decomposed into pairs of pants, and using the state-operator correspondence one understands the path integral on an arbitrary Riemann surface as the insertion of multiple resolutions of the identity. Using this gluing and sewing construction, one can write an arbitrary correlation function on a Riemann surface in terms of the spectrum of operators and their OPE coefficients.

In higher dimensions, such a gluing and sewing construction fails, since not all manifolds can be obtained this way. Furthermore, a state-operator correspondence is only known for states on the sphere, using a conformal transformation between $\mathbf{R}^d$ and the cylinder $S^{d-1} \times \mathbf{R}$. This presents us with an important difficulty when trying to understand constraints that CFTs should obey on more general manifolds. For example, consider a spatial manifold $\mathcal{M}_{d-1}$. The partition function

$$Z(\beta) = \mathrm{Tr}_{\mathcal{H}_{\mathcal{M}_{d-1}}} e^{-\beta H} = Z(\mathcal{M}_{d-1} \times S^1_\beta), \tag{1.1}$$

where $\mathcal{H}_{\mathcal{M}_{d-1}}$ is the Hilbert space on the given spatial manifold. Conformal field theories should still be modular invariant in higher dimensions, where by modular invariant we mean that at least their partition functions should be invariant under the action of the mapping class group of the manifold. One can obtain interesting results from this constraint, in particular when the manifold $\mathcal{M}_{d-1}$ contains circles. For example, one can obtain a generalized Cardy formula [8] or place constraints on holographic CFTs [9] (see also [10, 11] for similar ideas).

However, even when there are circles, modular invariance only relates states in the Hilbert space on the spatial manifold $\mathcal{M}_{d-1}$. The absence of a state-operator correspondence prevents us from relating modular invariance to properties of operators of the theory. The goal of this paper is to address this issue and investigate whether one can formulate a state-operator correspondence on other manifolds than $S^{d-1}$. For concreteness, we will focus on $d = 3$ and take the spatial manifold to be a torus $\mathbf{T}^2$.[1]

We start off in section 2 by reviewing the state-operator correspondence for states on $S^{d-1}$

---

[1]If the theory contains fermions, we will always assume anti-periodic boundary conditions for the fermions.

and discuss a natural generalization to the torus which sets the stage for the rest of the paper. In particular, we argue that a state-operator map can only exist if there is a compact manifold which prepares the ground state. Excited states are obtained by the addition of operators in the path integral and for the torus they are more naturally associated to line operators. Focussing on the ground state, we study a particular example of a compact Euclidean manifold with $\mathbf{T}^2$ boundary in section 3: a hemisphere times a circle. We show that such a path integral does not in general prepare the ground state, by explicitly showing a mismatch for the free boson and holographic CFTs.

In section 4, we generalize our discussion to arbitrary three-manifolds $M$ with a boundary two-torus. We connect local extrema of the energy functional to the existence of a conformal Killing vector on $M$. In particular, we conjecture that a manifold $M$ with a boundary two-torus prepares the ground state if and only if the manifold possesses a conformal Killing vector normal to the boundary two-torus. We argue that the only manifold that achieves this is the product of $M$ and a semi-infinite line. This provides a CFT analog of the Horowitz-Myers conjecture for the AdS-Soliton. Our arguments do not rely specifically on having a two-torus and apply equally well to other manifolds which are not spheres. We finish with a discussion in section 5.

## 2 The state-operator correspondence

The usual state-operator map is a powerful tool in conformal field theory. It relates

$$\text{Local operator on } \mathbf{R}^d \qquad \Longleftrightarrow \qquad \text{State on } S^{d-1}. \tag{2.2}$$

This relation exists because of a conformal transformation from $\mathbf{R}^d$ to $\mathbf{R} \times S^{d-1}$. Consider the Euclidean plane $\mathbf{R}^d$ whose metric we write in spherical coordinates

$$ds^2 = dr^2 + r^2 d\Omega_{d-1}^2. \tag{2.3}$$

If now do the coordinate change $r = e^{t_E}$ we get

$$ds^2 = e^{2t_E}(dt_E^2 + d\Omega_{d-1}^2), \tag{2.4}$$

which is the metric on the cylinder $\mathbf{R} \times S^{d-1}$ once we remove the conformal factor $\Omega^2 = e^{2t_E}$. Under this map, the dilatation operator $D = r\partial_r$ gets mapped to the operator $\partial_{t_E} = H$, the Hamiltonian on the sphere. Eigenstates of this Hamiltonian get mapped to operators that transform appropriately under scaling, namely local operators.

To be more precise, we have a Euclidean cylinder geometry that is preparing a state at $t_E = 0$, and the state is specified by a choice of boundary conditions at $t_E = -\infty$ (see Fig. 1). Using the exponential map, one can see from the plane point of view that the state is prepared on the sphere sitting at $r = 1$, and the choice of boundary condition at $t_E = -\infty$ gets mapped to a local condition at $r = 0$. The power of the state-operator correspondence is that the condition at $r = 0$ is simply the choice of insertion of a local operator of the CFT and moreover that these operators are in one-to-one correspondence with the states of the Hilbert space on $S^{d-1}$. Furthermore, the manifold over which the path integral is done has become compact. One can think of this Euclidean manifold as a filling of the spatial manifold on which the state is prepared. Because the manifold is compact, the non-local boundary condition at $t_E = -\infty$ has been mapped to a single-point. We will take these two properties (a bijective map and a compact manifold) to be fundamental properties of any state-operator correspondence and investigate whether there exists generalizations to other manifolds[2].

---

[2]Here we will be considering states on a compact manifold. If we were interested instead in states of a theory on a non-compact manifold like $\mathbf{R} \times S^1$, the Euclidean manifold would obviously need to be non-compact in the infinite spatial direction.

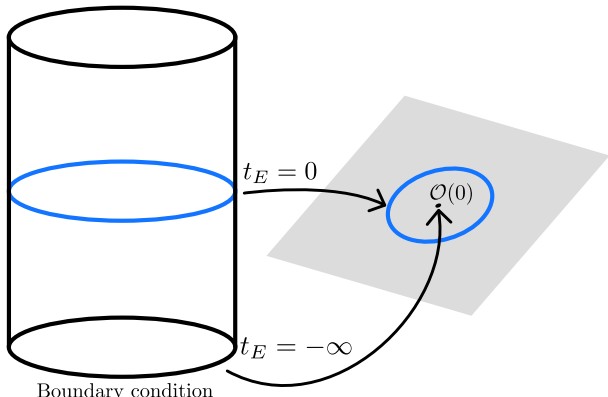

Fig. 1: The state operator correspondence

We would like to emphasize that having a compact manifold is really a necessary condition for a state-operator correspondence to exist. It is obviously true that one can obtain generic states in the Hilbert space by considering a Euclidean path integral over a semi-infinite line once we pick appropriate boundary conditions at $t_E = -\infty$. These boundary conditions need to be specified at every single point along the spatial manifold. A state-operator correspondence is a much stronger statement: it can only emerge when that boundary condition is replaced by a new boundary condition on a manifold of lower dimension. In the case of the usual state operator correspondence, it replaces a boundary condition on $S^{d-1}$ to that at a point. In this paper, the codimensionality of the operator will be different but it follows the same general principle.

For the sphere, the vacuum is prepared without the insertion of any operator. In particular, the vacuum is associated to the identity operator. It appears natural to postulate that in any generalization, the vacuum should be prepared by a compact Euclidean path integral without the insertion of any operator, and that excited states are built on top of the vacuum by adding operators in the path integral. For now, we will take this as an assumption and we will discuss the validity of this hypothesis in section 4. On general grounds, it is interesting to understand the set of states that can be generated by compact Euclidean path integral, particularly in the context of holography where the states in the gravitational dual are the no-boundary states à la Hartle-Hawking [12]. In the end, whichever family of path integrals we propose, it must at least be able to produce the vacuum state. If not, the bijective map breaks down and the state operator correspondence with it.

## 2.1 A state-operator correspondence on the torus?

We would like to understand how to generalize this notion of a state-operator correspondence to other spatial manifolds than $S^{d-1}$. In this paper, we will consider a particularly simple choice: the torus $\mathbf{T}^{d-1}$. For simplicity, we will set $d = 3$ and consider CFTs in three spacetime dimensions although the generalization to higher dimensions is straightforward.

The spatial manifold on which we want to prepare a state at $t_E = 0$ will be $\mathbf{T}^2 = S^1_{L_1} \times S^1_{L_2}$ where $2\pi L_{1,2}$ are the lengths of the two circles. We wish to understand the relation between states on this spatial manifold and the operator content of the theory.

States can be prepared by doing a path integral over a Euclidean manifold $M$ with boundary $\Sigma$. The boundary of $M$ is precisely the spatial slice of the theory on which the state lives. In our case the boundary is a two-torus. Different states are prepared by picking different manifolds $M$, or by the insertion of operators on any given $M$. They need not necessarily be local as we will see shortly. We emphasize again that we want the Euclidean manifold to be compact if

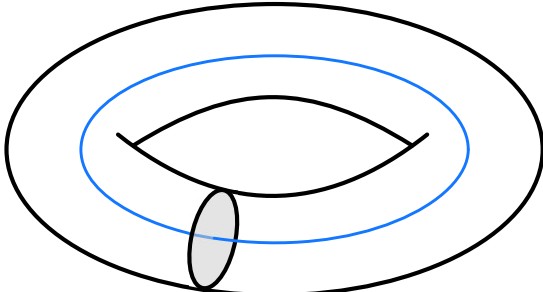

Fig. 2: A picture of the potential line operator/state correspondence. The state is prepared on the boundary of the donut: the two-torus. We can path integrate inwards until we reach the center of the donut where the operator is inserted (the blue line).

there is any hope to associate states to operators.

A simple example of such a manifold $M$ is the disk times a circle. This geometry is that of a donut and we can path integrate over it, with or without the insertion of operators, to produce a state on the two-torus.

The first observation is that states on the torus will more naturally map to non-local operators, contrarily to what happened on the sphere. To see this, imagine defining a state on the spatial torus. One can then path integrate inwards towards the "jelly" of the donut. Once we reach the center of the disk, we obtain some type of boundary condition there. This is depicted in Fig. 2. One can also work in the opposite direction: set a boundary condition at the center of the disk and path integrate outwards. This will yield a state on the spatial torus. It seems natural to associate the boundary conditions at the center of the disk to line operators: these operators are non-local, since they extend in the direction of the other circle, see Fig. 2. Since these line operators are defined by some local boundary condition for the fields of the theory, one can in principle use the same boundary conditions also to define line operators in flat space. It would be interesting to explore the properties of these line operators in more detail.

## 2.2 Vacuum state

The first question to address is whether the map described above is really bijective. Can we produce an arbitrary state of the theory by inserting the appropriate line operator?[3] We will address an easier version of this question: can we at least produce the vacuum state in such a manner? It is very tempting to associate this state to a path integral with no operator insertions, but with what filling of the two-torus?

Let us first discuss the usual way of preparing the vacuum in a QFT. If we consider the thermal partition function which is a path integral $\mathbf{T}^2 \times S^1$,

$$Z(\beta) = \mathrm{Tr}_{\mathbf{T}^2} \, e^{-\beta H}, \tag{2.5}$$

where $\beta$ is the periodicity of the $S^1$, then we can isolate the contribution from the ground state by taking the zero temperature limit of $Z(\beta)$. This means that $\beta$ goes to infinity and hence the path integral is done over $\mathbf{T}^2 \times \mathbf{R}$. The semi-infinite cylinder therefore always prepare the vacuum in any QFT but it is non-compact. The question we would like to address is whether

---

[3]One could also in principle imagine inserting additional local operators elsewhere on $M$. Ideally, one would hope that such operators don't give additional states in the theory and that one can reduce their insertion to different line operators at the jelly by using some type of OPE between the local and line operators. It would be interesting to pursue this idea further but it will not be important in this work.

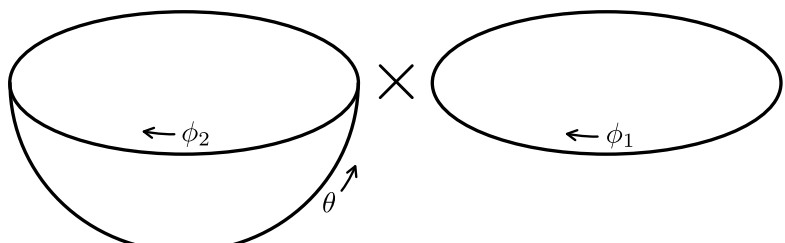

Fig. 3: The manifold over which we path integrate, with the two circles forming the two-torus on which the state is prepared.

this is the only manifold that can achieve this. If we replace the torus by a sphere, we know that the semi-infinite cylinder is not the only manifold: there is also the half three-sphere. This occurs because the two manifolds are conformally related and is the essence of the state-operator correspondence. We would like to find a similar setting for the torus states.

In the next section, we will warm up by discussing a particular example of a compact Euclidean manifold very similar to the one described above. Instead of taking a disc times a circle, we will consider a hemisphere times a circle. Usually, we want to have a Euclidean section on which we can smoothly glue a Lorentzian manifold to perform time evolution. A disc times a circle will result in a singularity at the junction, which will possibly introduce sources and/or singularities in the theory. A hemisphere on the other hand glues on smoothly in the sense that the metric will have continuous first derivatives. For the usual state-operator correspondence on the sphere, the disk and the hemisphere are related by a conformal transformation. This is no longer true on the torus, which leads us to directly consider the hemisphere filling.

## 3 An example of a compact Euclidean filling: the hemisphere

As explained in the previous section, this filling is a natural candidate to consider since it is compact and is very similar to the manifold that we know works for states on the sphere. Furthermore, out of all compact manifolds that fill the torus it is the one with the most symmetry. If the state-operator correspondence is to work, it seems logical to think that the path integral without the insertion of any operators produces the vacuum, and that the excited states are obtained by adding additional operators (local or not). We thus need to check whether our manifold produces the ground state.

Let us parametrize the filling by three angular coordinates $(\phi_1, \theta, \phi_2)$, as shown in Fig. 3 and refer to this manifold as $M_h$. Suppose $|\psi\rangle$ is the state we prepare by doing a path integral over $M_h$. A simple way to check whether $|\psi\rangle$ will be the vacuum is to compare its energy to the ground state energy. With the chosen coordinates, the metric on $M_h$ is

$$ds^2 = L_1^2 d\phi_1^2 + L_2^2 \left(d\theta^2 + \sin^2\theta \, d\phi_2^2\right), \tag{3.6}$$

where we $\phi_i$ are angular coordinates running from 0 to $2\pi$, $\theta$ will take values from 0 to $\pi$. The energy of the state $|\psi\rangle$ is

$$E_{|\psi\rangle} = -L_1 L_2 \int d\phi_1 d\phi_2 \, \xi^\mu n^\nu \langle \psi | T_{\mu\nu} | \psi \rangle, \tag{3.7}$$

where the minus sign comes from the fact that we are evaluating a Euclidean correlator whereas the energy is defined by integrating the stress tensor in Lorentzian signature. The

vector $\xi^\mu$ is a unit vector pointing in the direction in which we want to generate time translations and $n^\mu$ is a unit vector orthogonal to the spatial slice. In this case, they are equal and read

$$n^\mu = \xi^\mu = \frac{\partial_\theta}{L_2}. \tag{3.8}$$

Note that the bra state is obtained by path integrating over the north pole down to the equator. The expectation value is then taken by computing the one-point function of the stress-tensor on the manifold $S^1 \times S^2$. The energy then becomes

$$E_{|\psi\rangle} = -\frac{L_1}{L_2} \int d\phi_1 d\phi_2 \, \langle\psi|T_{\theta\theta}(\phi_1, \pi/2, \phi_2)|\psi\rangle. \tag{3.9}$$

We will now make the reasonable assumption that $|\psi\rangle$ does not break translational symmetry along the $S^1_{L_1}$, nor does it break the rotational invariance along the $\phi_2$ direction of the sphere. The integrals in equation (3.9) are then easily done and we obtain

$$E_{|\psi\rangle} = -4\pi^2 \frac{L_1}{L_2} \langle T_{\theta\theta}(0, \pi/2, 0)\rangle. \tag{3.10}$$

We can even go a step further by assuming rotational invariance on the sphere which gives

$$T_{\theta\theta}(0, \pi/2, 0) = T_{\phi_2\phi_2}(0, \pi/2, 0). \tag{3.11}$$

Using tracelessness of the stress tensor,

$$T_{\theta\theta} + \frac{1}{\sin^2\theta} T_{\phi_2\phi_2} + \frac{L_2^2}{L_1^2} T_{\phi_1\phi_1} = 0, \tag{3.12}$$

we can write

$$\langle T_{\theta\theta}(0, \pi/2, 0)\rangle = -\frac{1}{2}\frac{L_2^2}{L_1^2} \langle T_{\phi_1\phi_1}(0, \pi/2, 0)\rangle. \tag{3.13}$$

Moreover, the energy of the state $|\psi\rangle$ can also be interpreted in terms of a thermal energy on $S^2 \times S^1$

$$E_{\text{th}} = -4\pi \frac{L_2^2}{L_1^2} \langle T_{\tau\tau}(0, \pi/2, 0)\rangle, \tag{3.14}$$

where we have replaced $\phi_1$ by the imaginary time coordinate $\tau$ to make this more transparent. This allows us to rewrite (3.10) as

$$E_{|\psi\rangle} = -\frac{\pi}{2}\frac{L_1}{L_2} E_{\text{th}}. \tag{3.15}$$

We can actually build two different states, depending on which cycle we make contractable. In what follows, it will be more convenient to make the larger circle contractible but we can always consider either case. So far, we have connected the expectation value of the energy in a particular state to a thermal expectation value on the sphere. It is quite complicated to evaluate such an expression in general, since it is highly theory dependent. The Casimir energy on the torus is also highly theory dependent [9] so there is still hope that they could match.

One important comment is that the thermal energy is necessarily positive in three dimensions since the vacuum energy on the sphere vanishes. This means the energy we have constructed is negative which is good since the Casimir energy on the torus is always negative. Whether it is negative enough to match the actual vacuum energy is still unclear at this point. To gain some insight, we will evaluate (3.15) in two examples: holographic CFTs and the free boson. We are after a general state-operator correspondence valid in any CFT so we can test it in particular examples. Finding a single counter-example would be enough to invalidate the proposal for the manifold $M_h$.

### 3.1 Holographic CFTs

For holographic field theories it is straightforward to check whether the state we prepared has a higher energy than the ground state energy on $\mathbf{T}^2$. We will assume the CFT to be dual to Einstein gravity. Starting with the ground state energy on the torus, the relevant solution to the equations of motion is the AdS-Soliton [13]. The metric is

$$ds^2 = -\frac{r^2}{\ell_{AdS}^2}dt^2 + \frac{\ell_{AdS}^2 dr^2}{r^2(1-(L_1/r)^3)} + r^2\left(1-(L_1/r)^3\right)dx_1^2 + \frac{r^2}{\ell_{AdS}^2}dx_2^2. \tag{3.16}$$

This geometry is the dominant solution for $L_1 < L_2$. We can easily extract the Casimir energy:

$$E_0(L_1, L_2) = -\frac{2\pi\ell_{AdS}^2}{27G_N}\frac{L_1 L_2}{L_1^3}. \tag{3.17}$$

The factor of $\ell_{AdS}^2/G_N$ means the energy is of order $N^2$.

We now compare this result to the thermal energy of the state $|\psi\rangle$ on $S_{L_2}^2$. This is equivalent to computing the energy of either an AdS-Schwarzschild black hole or thermal AdS, depending on the value of $L_1/L_2$. If the size of $1/L_1$ is smaller then the Hawking-Page temperature, we are dealing with thermal AdS in the bulk which has energy of order $\mathcal{O}(1)$. This can never give the ground state energy as it doesn't scale with $N$. We are thus left with states prepared on a torus with $1/L_1$ larger then $T_{HP}$. Their energy is given by the energy of a black hole in $AdS_4$. The metric of such a black hole is given by

$$ds^2 = f(r)d\tau^2 + \frac{dr^2}{f(r)} + r^2 d\Omega_2, \tag{3.18}$$

where

$$f(r) = 1 + \left(\frac{r}{\ell_{AdS}}\right)^2 - \frac{2MG_N}{r}. \tag{3.19}$$

This black hole has mass $M$, which is related to the thermal energy on $S_{L_2}^2$ as $E_{\text{th}} = \frac{\ell_{AdS}}{L_2}M$. The goal is now to express $M$ in terms of the periodicity of $\tau$. This is most easily done by first solving $2\pi L_1 = 4\pi/f'(r)\big|_{r=r_h}$ in terms of $r_h$ and then using that in the expression for $M$ as a function of $r_h$. To compare, we form the ratio between the Casimir energy in equation (3.17) and the energy of the state $|\psi\rangle$ given in (3.15):

$$R(\alpha) = \frac{E_{|\psi\rangle}}{E_0} = \frac{1}{4}\left(2 + 2\sqrt{1-3\alpha^2} + 3\alpha^2\sqrt{1-3\alpha^2}\right), \tag{3.20}$$

where $L_1 = \alpha L_2$. This function is bounded from above by 1, see Fig. 4(a), proving that the Casimir energy is always lower then the energy of the state we are preparing. We can get very close to the ground state by going to very high temperatures (small $L_1$), but we will never reach it without pinching off the torus.

### 3.2 Free boson

We can repeat the above analysis for a massless (but conformally coupled) free boson. Equation (3.15) instructs us to compute the expectation value of the energy for a free boson on the 2-sphere with radius $L_2$ at inverse temperature $2\pi L_1$. To do so, we first compute the partition function. It is given by [14]

$$Z = \prod_{l=0}^{\infty}\left(\frac{1}{1-e^{-2\pi L_1(l+1/2)/L_2}}\right)^{2l+1}, \tag{3.21}$$

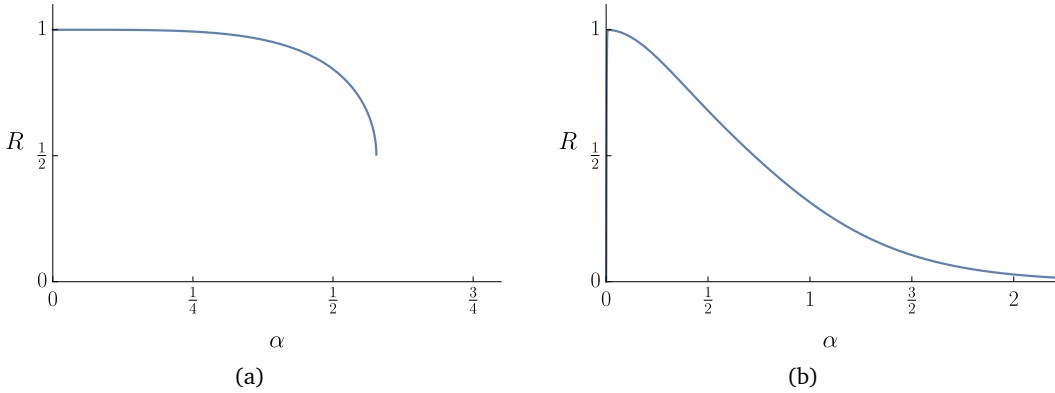

Fig. 4: Ratios $R$ plotted as a function of the ratio of the two circle lengths $\alpha = L_1/L_2$ for (a) a Holographic CFT, (3.20), and (b) for a free boson where we summed (3.22) up to $l = 200$. The fact that $R$ shoots up near $\alpha = 0$ in (b) is a consequence of this finite sum.

with $l$ the orbital angular momentum. The energy is given by $E_{\text{th}} = -\frac{1}{2\pi} \partial_{L_1} \log Z$ and so we have

$$E_{\text{th}} = \frac{1}{2L_2} \sum_{l=0}^{\infty} (2l+1)^2 \frac{e^{-2\pi L_1(l+1/2)/L_2}}{1 - e^{-2\pi L_1(l+1/2)/L_2}}. \tag{3.22}$$

We should compare this energy to the Casimir energy of the free boson on the torus. This was calculated, for example in [15] and reads [4]

$$E_0(L_1, L_2) = -L_1 L_2 \pi \left( \frac{\zeta(3)}{4\pi^3 L_1^3} + \frac{1}{12\pi L_1 L_2^2} + \frac{2}{\pi^2 L_1^2 L_2} \sum_{m,n=1}^{\infty} \frac{n}{m} K_1 \left( \frac{2\pi n m L_2}{L_1} \right) \right), \tag{3.23}$$

where $K_1$ is the modified Bessel function of the second kind. We can evaluate the ratio $R = \frac{E_{|\psi\rangle}}{E_0}$ numerically, which results in Fig. 4(b). We see that the ratio is always smaller than one. It only approaches unity in the infinite temperature limit as in the holographic example. In fact, we can see that the two agree at infinite temperature by calculating the expectation value of the energy exactly in the limit of infinite temperature. Replacing the sum in (3.22) by an integral, we obtain $E_{\text{th}} = \frac{\zeta(3)L_2^2}{2\pi^3 L_1^3}$ which matches the Casimir energy in the limit of infinite $L_2$ once we apply (3.15).

Our examples clearly demonstrate that the state prepared by the Euclidean path integral over the hemisphere times a circle does not in general prepare the ground state in a CFT. We have established this result by considering two explicit counter-examples. One may ask whether the path integral without any operator insertions is really the right thing to do to prepare the vacuum. This was the case for states on the sphere but it could turn out to be different here. We will comment on this issue in the discussion section.

Nevertheless, the example studied in this section should illustrate some obstructions to constructing a state-operator correspondence for the torus. One may of course wonder whether this resulted from making a poor choice of compact Euclidean manifold. We picked the manifold satisfying our criteria with the most symmetry, but perhaps a different manifold can produce the vacuum. In order to disprove the existence of a state-operator correspondence, one cannot restrict oneself to a single compact Euclidean manifold. In the following section, we explore more general compact manifolds and see how some of the results given above generalize.

---

[4]This expression is symmetric under the exchange of $L_1$ and $L_2$, but only implicitly so.

# 4 General Manifolds

In order to disprove the existence of a state-operator map for CFTs on a torus, it is enough to show that there is no compact Euclidean manifold $M$ with boundary $\partial M = \mathbf{T}^2$ that prepares the ground state of the theory. If a manifold can produce the ground state, it means that it extremizes the energy functional. When $M$ becomes $\mathbf{R}_+ \times \mathbf{T}^2$, the energy functional must reach a global minimum, since this manifold prepares the ground state in any QFT. The question now is whether other manifolds can also prepare this state. In this section, we tie this question to the existence of a conformal killing vector for the manifold $M$. We will argue that a manifold extremizes the energy functional if and only if the manifold $M$ possesses a conformal killing vector normal to the two-torus. In this section, we prove one direction of the statement and give evidence for the other.

## 4.1 Trying to Prepare the Vacuum with General 3-manifolds

Consider an arbitrary Euclidean manifold $M$ with boundary $\partial M = \Sigma = \mathbf{T}^2$ on which the CFT state lives[5]. The energy functional is given by

$$\mathcal{E}[M] = \int_\Sigma d\Sigma^\nu \xi^\mu \langle T_{\mu\nu} \rangle. \tag{4.24}$$

This quantity depends significantly on $M$ since the choice of $M$ selects a particular (ket) state in which the stress tensor one-point function is evaluated. The bra state is prepared using the orientation reversed version of $M$, denoted by $\overline{M}$. This manifold is glued to $M$ along $\Sigma$. The vector $\xi$ is normal to $\mathbf{T}^2$, and points in the direction in which we want to generate time translations. We want the vector $\xi^\mu$ to be normal to the two-torus as well in order to assure a smooth gluing to a Lorentzian manifold. As before, we will assume that the manifold can be glued to the Lorentzian manifold $\Sigma \times \mathbf{R}_+$ in such a way that the metric has continuous first derivatives.

We would like to investigate the behaviour of this functional when we insert additional operators in the path integrals by turning on a source $h$ for the operator $\mathcal{O}$ as

$$\exp\left( \epsilon \int_M d^3x \sqrt{g} h(x) \mathcal{O}(x) \right). \tag{4.25}$$

Keeping the linear order in $\epsilon$, this provides a set of first order variations of the energy functional. Note that the shape variations of $M$ are also included in (4.25) as they correspond to insertions of the stress tensor.

A necessary condition for a state to be the vacuum is that it minimizes the energy functional (4.24). For it to be a local minimum, the first order variations of (4.24) need to vanish. The first order variations of (4.24) have two contributions, one coming from the bra state where $\mathcal{O}$ is inserted on $\overline{M}$ and another where it is inserted on $M$. Since the sources are real and the set-up has a reflection symmetry through $\Sigma$, both contributions are equal so we can consider only sources inserted on $M$ and multiply the result by a factor of 2. We thus obtain

$$\delta E = 2\epsilon \int_\Sigma d\Sigma^\nu \int_M d^3x \sqrt{g} \xi^\mu h(x) \langle T_{\mu\nu}(y) \mathcal{O}(x) \rangle_{\text{con}}. \tag{4.26}$$

In the above expression, $y$ is a coordinate on $\Sigma$ and only the connected part of the correlation function remains. The condition that the first order variations of the state need to vanish requires the integrated correlation function above to vanish.

---

[5]The generalization to other manifolds than $\mathbf{T}^2$ and other dimensions is straightforward.

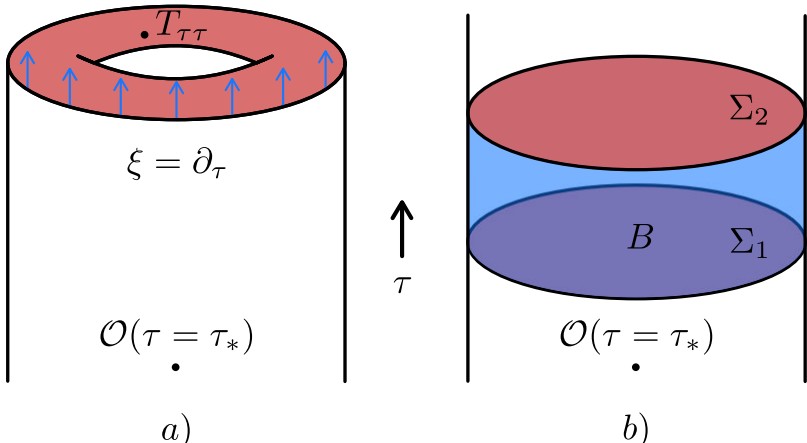

Fig. 5: a) Manifold $M = \mathbf{R}_+ \times \mathbf{T}^2$, which prepares the ground state. This manifold has a Killing vector $\xi$ along the $\tau$ direction. The stress tensor $T_{\tau\tau}$ is integrated over the torus (red shaded region). To consider the effect of an operator insertion, we have inserted one at $\tau = \tau_*$. b) We can deform the region of integration of $T_{\tau\tau}$ from $\Sigma_1$ to $\Sigma_2$. We pick up a bulk contribution from integration over the blue region $B$ between $\Sigma_1$ and $\Sigma_2$. This contribution vanishes when there is a conformal Killing vector.

It seems impossible to get them to vanish in an arbitrary CFT for arbitrary operator $\mathcal{O}$ and source $h$ without a symmetry argument. In other words, it seems too much to ask that they vanish for dynamical reasons, so it must come from a kinematical argument tied to $M$. Can we find a manifold that achieves this? Let us start with the manifold that we know for sure prepares the vacuum, namely $\mathbf{R}_+ \times \mathbf{T}^2$. In this case, the vector $\xi$ is simply the generator of time translations $\partial_\tau$, where $\tau$ is the direction along $\mathbf{R}$. In this case, $\xi$ is a conformal killing vector which has significant consequences. Consider an operator insertion at a time $\tau = \tau_*$ (with delta function source) while the stress tensor is inserted at $\tau = 0$, see Fig. 5.

Since $\xi$ is a Killing vector, we can move the position of the stress-tensor to whichever slice we want in $M$. To see this, consider

$$
\begin{aligned}
\delta E & \equiv 2\epsilon \int_{\Sigma_1} d\Sigma^\nu \xi^\mu \langle T_{\mu\nu} \mathcal{O}(\tau = \tau^*) \rangle_{\text{con}} \\
& = 2\epsilon \int_{\Sigma_2} \xi^\mu \langle T_{\mu\nu} \mathcal{O}(\tau = \tau^*) \rangle_c \, d\Sigma^\nu - 2\epsilon \int_B d^3 x \, \nabla^\nu \big( \xi^\mu \langle T_{\mu\nu} \mathcal{O}(\tau = \tau^*) \rangle_c \big).
\end{aligned} \tag{4.27}
$$

The second term vanishes if $\xi$ is conformal Killing, by tracelessness and conservation of the stress-tensor [6]. This means we can move the position of the stress-tensor to any $\Sigma_2$ at no cost. In particular, we can deform the contour to the part of the manifold where no operator is inserted, namely $\bar{M}$. Recall that there were two terms in the first order variation, one coming from an operator inserted in $M$ and one from $\overline{M}$. Since they are equal, we have written the total contribution as twice that of the contribution where the operator is in $M$. Since no operator is inserted in $\overline{M}$, we can move the stress-tensor arbitrarily far away from the operator $\mathcal{O}$ where the contribution gives $\delta E = 0$.

---

[6]This is the only part of our arguments that cares about the dimension. In odd dimensions, there is no anomaly. In even dimensions, we could pick up the contribution from the anomaly. However, this will make it even harder to get the variations to vanish if anything so we don't believe that is particularly important.

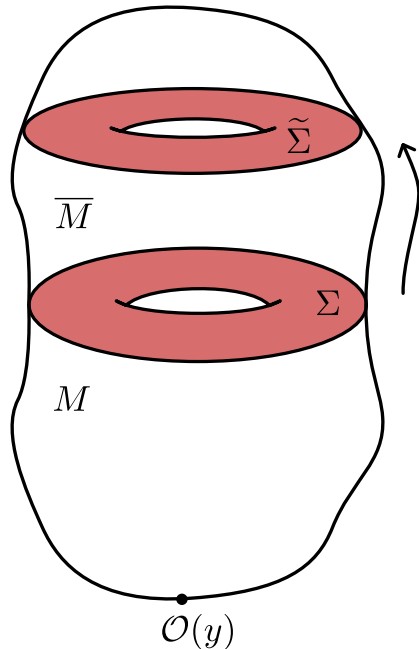

Fig. 6: We can deform the contour from $\Sigma$ to $\tilde{\Sigma}$ at no extra cost. If the manifold is compact, we can deform it to the tip where it shrinks to zero.

## 4.2 Conformal killing vector implies local extremum

Now consider a general Euclidean manifold. We will first show that if the manifold possesses a conformal killing vector normal to $\Sigma$, then all the first order variations vanish. The proof goes as follows: a conformal killing vector allows us to deform the contour at no cost, following the flow set by the vector. If the manifold is non-compact, we can always deform the contour arbitrary far away which makes the first order variation vanish. If the manifold is compact and smooth, some cycle must shrink to zero-size somewhere. By deforming the contour to that point, we can get the first term in (4.27) to vanish. This is illustrated in Fig. 6. The existence of a conformal Killing vector provides us with a symmetry argument of why the correlators should vanish. In terms of Killing vectors that satisfy our boundary condition, the manifold $M = \mathbf{R}_+ \times \mathbf{T}^2$ is the only possible choice. If the vector is only conformal Killing, there is more room. The question becomes understanding which three-manifolds with a $\mathbf{T}^2$ boundary have a conformal Killing vector with the appropriate boundary conditions.

Before going further, it is important to specify precisely what we mean with shrinking the contour to zero size. There are two options: either we shrink a one-cycle along a one-dimensional manifold (like for the hemisphere state) or we shrink a two-cycle. In mathematics, the formal notion of shrinking one-cycles in the manifold $M$ with boundary $\Sigma$ is compressibility and $M = S^1 \times D^2$ is the only topology that has a compressible boundary torus where one of the torus cycles shrinks to zero. All other topologies have an incompressible boundary and are called Haken manifolds. We will now show that manifolds with topology $S^1 \times D^2$ cannot have a conformal Killing vector normal to the boundary.

First, we show that only two-cycles can shrink along a conformal Killing vector flow. The place where any cycle shrinks must be a fixed point of the conformal Killing vector flow. Locally, the manifold $M$ is $\mathbf{R}^3$ which has an $SO(4,1)$ worth of conformal Killing vectors. The only ones that have converging field lines are the vector fields corresponding to scale and special conformal transformations. The fixed points of those flows are always isolated points. They can never be submanifolds. To see this more clearly, imagine trying to shrink a one-cycle along

a conformal Killing vector flow. Near the point where the cycle shrinks the metric is

$$ds^2 = dx_{\parallel}^2 + dr^2 + r^2 d\phi^2, \tag{4.28}$$

where $x_{\parallel}$ is a non-shrinking direction. Moving along the shrinking direction $r$ is not a conformal Killing direction. If we try to make a conformal transformation, we find

$$ds^2 = r^2 \left( \frac{dx_{\parallel}^2}{r^2} + \frac{dr^2}{r^2} + d\phi^2 \right) = e^{2\xi} \left( \frac{dx_{\parallel}^2}{e^{2\xi}} + d\xi^2 + d\phi^2 \right), \tag{4.29}$$

for $r = e^{\xi}$. The direction $\xi$ is only a conformal Killing direction if we get rid of $x_{\parallel}$ and replace the circle by a two-sphere.

Having seen that the only way to shrink cycles along a conformal Killing direction is with two-spheres, we can now show that there are no compact manifolds with a conformal Killing vector orthogonal to $\Sigma$. This follows from the fact that smooth conformal Killing vector flows cannot change topology. Since the topology on $\Sigma$ is that of a torus, we cannot smoothly deform it to a two-sphere that shrinks to zero-size along a conformal Killing vector flow as that would alter the Euler characteristic of $\Sigma$. This completes the proof. The only manifold that has an appropriate conformal Killing vector is therefore $M = \mathbf{R}_+ \times \mathbf{T}^2$, or manifolds in the same conformal class.

Thus far, we have proved that given a manifold $M$ with an appropriate conformal Killing vector, all first order variations vanish and we have a local extremum of the energy functional. Furthermore, we showed that this cannot be accomplished on a compact manifold. In order to prove that the vacuum cannot be obtained from a compact manifold, we would still need to establish the converse, namely that a local extremum implies the existence of a conformal Killing vector. We were unfortunately not able to prove this, but we give heuristic argument in the following subsection.

### 4.3 Local extremum implies existence of a conformal Killing vector?

We would now like to prove the converse, namely that if all first order variations vanish, then the compact manifold $M$ must have a conformal killing vector with the appropriate boundary conditions. Our starting point is therefore that for the state prepared by $M$ we have

$$\int_{\Sigma} d\Sigma^{\nu} \xi^{\mu} \langle T_{\mu\nu}(x) \mathcal{O}(y) \rangle_{\text{con}} = 0 \tag{4.30}$$

for all $\mathcal{O}$ and all insertion points. We also assume that $\langle T_{\mu\nu}(x) \mathcal{O}(y) \rangle_{\text{con}} \neq 0$ for any operator $\mathcal{O}$ irrespective of its insertion point. In other words, the reason why the integrated two point function vanishes should have a purely kinematical origin rather than dynamical. It is possible that in a particular CFT, these two point functions vanish for arbitrary operator $\mathcal{O}$ and arbitrary insertion points. Although extremely unlikely, we have not been able to prove it so it remains logically possible. We will therefore take it to be an assumption. Even with this assumption, we will not be able to formally prove the existence of a conformal Killing vector, but we will give some heuristic arguments.

The first argument goes as follows. Extend the vector field $\xi^{\mu}$ to an arbitrary vector field $\hat{\xi}^{\mu}$ on $M$, see Fig. 7, and then deform the contour to the point where a cycle shrinks. We can use Stokes theorem to convert the integral in (4.30) to an integral over $M$,

$$0 = \int_{\Sigma} d\Sigma^{\nu} \xi^{\mu} \langle T_{\mu\nu}(x) \mathcal{O}(y) \rangle_{\text{con}} = \int_{M} d^3 x \sqrt{g} (\nabla^{\mu} \hat{\xi}^{\nu}) \langle T_{\mu\nu}(x) \mathcal{O}(y) \rangle_{\text{con}}, \tag{4.31}$$

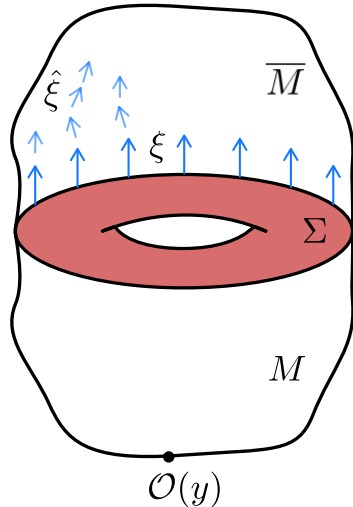

Fig. 7: Compact manifold $M$ which prepares a state $|\psi\rangle$ on $\partial M = \mathbf{T}^2$. The vector field $\xi$ normal to $\mathbf{T}^2$ is extended to a vector field $\hat{\xi}$ on $\overline{M}$. To compute the energy, the expectation value of the stress tensor $T_{\tau\tau}$ in the state $|\psi\rangle$ is integrated over $\mathbf{T}^2$ (red shaded region). To consider the effect of an operator insertion, we have inserted one at $y \in M$.

where we assumed $M$ to be compact. The term in round brackets can be symmetrized, which shows that if

$$\nabla^{(\mu}\hat{\xi}^{\nu)} = \alpha(x)g^{\mu\nu}, \tag{4.32}$$

for some $\alpha(x)$, equation (4.31) will be satisfied. We can fix $\alpha$ by taking the trace, which reduces (4.32) to the conformal Killing equation. However, using this to argue that a local extremum implies the existence of a conformal Killing vector is cheating. First of all, (4.32) is only a necessary condition if we can treat the two-point function as a variational function, i.e. $\langle T_{\mu\nu}(x)\mathcal{O}(y)\rangle_{\text{con}}$ needs to be an arbitrary function. Clearly this is not the case, because this correlator is both traceless and conserved. Second, we are rewriting $\delta E$ as a total derivative and when doing a variational principle these terms should be disregarded. This argument is therefore far from a convincing mathematical proof. Nevertheless, it is possible that a more sophisticated version of it could go further in proving the conjecture but we leave this for future work. We now turn to the second argument.

## 4.4 Gluing on a Lorentzian cylinder

Another argument why an extremum of the energy functional implies the existence of a conformal Killing vector goes as follows. Assume that we have found a manifold $M$ that prepares the ground state on $\mathbf{T}^2$. To time-evolve this state, we have to glue onto $M$ a Lorentzian manifold and ensure that the junction is smooth, meaning that normal derivatives agree. The ground state is an eigenstate of the Hamiltonian, so by denoting $t$ the time-coordinate along the Lorentzian manifold it has a time-translation symmetry $t \to t + a$. This symmetry holds for any correlation function computed in an energy eigenstate, see Fig. 8.

Near the junction of the Euclidean and Lorentzian manifolds, we still have the translation symmetry in the Lorentzian side. However, due to the smooth gluing condition, this symmetry should also be present on the Euclidean side. Moreover, if the state has a symmetry in the Lorentzian piece, upon analytic continuation, one might think that this symmetry should be preserved either in the form of a conformal Killing vector or just a Killing vector. For example, states on the sphere have this translation symmetry and in the Euclidean piece this symmetry

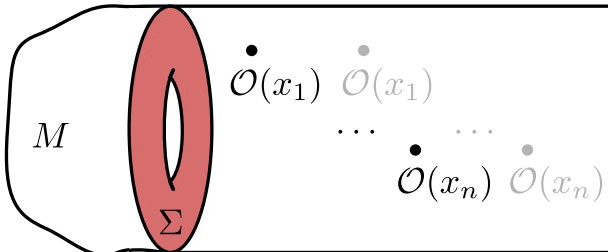

Fig. 8: A compact Euclidean cap $M$ prepares a state $|\psi\rangle$ on $\Sigma = \mathbf{T}^2$. A Lorentzian cyclinder $\mathbf{T}^2 \times \mathbf{R}_+$ is glued onto the cap to study time-evolution. If the state $|\psi\rangle$ is the vacuum (or more generally an energy eigenstate), correlation functions of operators inserted on the Lorentzian cyclinder exhibit time-translation symmetry. This symmetry should analytically continue on the cap $M$ as well.

is mapped to a conformal Killing vector. In particular in two dimensions, the conformal Killing vector is $\xi = -\sin(\theta)\partial_\theta$. Consequently, the Euclidean cap should also have a conformal Killing vector $\xi$ implementing the translation symmetry on the Euclidean side. The only manifold with such conformal vector fields are manifolds with topology $\mathbf{T}^2 \times \mathbf{R}$. Note that the arguments above also hold for any energy eigenstate. The line of reasoning here relies on an analytic continuation which makes it a very delicate argument and not a formal proof. It could also be the case that the symmetry is realized in a more complicated, not necessarily local way, on the Euclidean cap. It would be interesting to understand better consistency conditions on this analytic continuation of symmetries through the Euclidean/Lorentzian gluing. In the following subsection, we study the spectrum of a free conformally coupled scalar field theory which touches upon similar issues.

## 4.5 Spectral considerations

It is also instructive to analyze the problem for a free, conformally coupled scalar, from a spectral point of view. Consider the theory on $M^{d-1} \times \mathbf{R}$. We can canonically quantize the theory and construct the ground state wave functional. The action will be of the form

$$S = \int d^d x \sqrt{g}\, \phi(-\partial_\tau^2 + \nabla_{d-1})\phi, \tag{4.33}$$

where $\tau$ is the coordinate on $\mathbf{R}$ and $\nabla_{d-1}$ is time-independent. It consists of the Laplacian on $M^{d-1}$ plus a suitable coupling to the Ricci scalar curvature of $M^{d-1}$. It is not the conformal Laplacian on $M^{d-1}$ because the conformal coupling is the one appropriate for a $d$-dimensional theory, not a $d-1$-dimensional theory.

By canonical quantization, we find that the ground state wave-functional is determined by the spectrum and eigenfunctions of the operator $\nabla_{d-1}$. If we denote the eigenvalues by $E_i$ and the eigenfunctions by $\phi_i$, and introduce the inner product $\langle f, g \rangle = \int d^{d-1}x \sqrt{g}\, f^* g$ on $M$, then the unnormalized ground state wave-functional is

$$\Psi(\phi(x)) = \exp\left(-\frac{1}{2}\sum_i \sqrt{E_i}\langle \phi, \phi_i\rangle\langle \phi_i, \phi\rangle\right). \tag{4.34}$$

We can compare this to the wave function obtained by performing a path integral over a compact manifold $M^d$ whose boundary is $M^{d-1}$. To do this path integral we need to impose Dirichlet boundary conditions on the boundary, but it is easier to put sources on the boundary, compute the effective action for the sources, and then add $\int_{M^{d-1}} J\phi$ and integrate over the

sources $J$. The answer that one gets is that one has to restrict the propagator $G$ on $M^d$ to the boundary and then take the inverse of the propagator in the space of functions on the boundary. Thus

$$\Psi_{M^d}(\phi(x)) = \exp\left(-\frac{1}{2}\int d^{d-1}x\sqrt{g(x)}\int d^{d-1}y\sqrt{g(y)}\phi(x)(G(x,y))^{-1}\phi(y)\right). \quad (4.35)$$

We emphasize that in this expression we first need to restrict the propagator to the boundary and then compute its inverse, not the other way around. Comparing the two expressions, we see that it is very difficult to make the two equal to each other. They are equal if and only if

(i) There is a basis of eigenfunctions of the kinetic operator $\nabla_d$ on $M^d$ of the form $\phi^{i,k}$ with energy eigenvalues $E_{i,k}$ such that the restriction of $\phi^{i,k}$ to the boundary agrees with the eigenfunction $\phi_i$ of $\nabla_{d-1}$.

(ii) The relation

$$\frac{1}{\sqrt{E_i}} = \sum_k \frac{1}{E_{i,k}} \quad (4.36)$$

holds.

It is instructive to see how this works if $M_d = M_{d-1} \times \mathbf{R}$. Then the label $k$ is a continuous frequency $\omega$, and $E_{i,\omega} = E_i + \omega^2$. Equation (4.36) becomes the statement

$$\frac{1}{\pi}\int \frac{d\omega}{E+\omega^2} = \frac{1}{\sqrt{E}}. \quad (4.37)$$

The situation where $M_d$ is a half-sphere is a bit more complicated but still works. The conformal coupling in $d$ dimensions is $(d-2)/4(d-1)R$, and the Ricci curvature of a $d$-sphere is $d(d-1)$, so the conformal laplacian on $S^d$ has an extra additive piece equal to $(d-2)/4(d-1)\times d(d-1) = d(d-2)/4$, whereas the laplacian on the boundary $S^{d-1}$ receives an extra additive piece equal to $(d-2)/4(d-1)\times(d-1)(d-2) = (d-2)^2/4$. The eigenvalues of the ordinary (non-conformal) laplacian on the $d$-sphere are $k(k+d-1)$. Therefore the energy eigenvalues on $M^d$ are $k(k+d-1)+d(d-2)/4 = (k+d/2)(k+d/2-1)$. On the other hand, the eigenvalues of the kinetic term on $M^{d-1}$ are $k(k+d-2)+(d-2)^2/4 = (k+d/2-1)^2$. A spherical harmonic in $d$-dimensions labeled by $k$ contains a spherical harmonic in $d-1$-dimensions labeled by $k'$, when restricting to an equatorial plane, if $k \le k'$. Equation (4.36) therefore indeed holds and takes the form

$$\sum_{k=k'}^{\infty}\frac{1}{(k+d/2)(k+d/2-1)} = \sum_{k=k'}^{\infty}\frac{1}{k+d/2-1} - \frac{1}{k+d/2} = \frac{1}{k'+d/2-1}. \quad (4.38)$$

These two examples illustrate how fine-tuned the situation is. From this point of view it is also easy to independently verify that the example discussed in section 3 does indeed not work for a free conformally coupled scalar.

Already criterion (i) shows that the operator $\nabla_{d-1}$ can be extended to an operator which is defined over all of $M^d$ and which commutes with $\nabla_d$ by assigning eigenvalue $E_i$ to $\phi_{i,k}$. It is however not clear that this operator is local. If it were local we could probably argue for the existence of a conformal Killing vector. This is more or less the same obstruction as in the previous subsection, where it did strictly speaking not need to be the case that the Lorentzian time translation generator extends to a geometric vector field on the Euclidean cap.

To sum up, we gave give two general heuristic arguments why we believe our conjecture to be true and we gained additional insight by considering a conformally coupled scalar field. Unfortunately, a rigorous proof of our claim is still evading us. By specializing this time to holographic CFTs, we can in fact say a little more by relating our arguments to an old conjecture by Horowitz and Myers.

## 4.6 Holography and the Horowitz-Myers Conjecture

For generic conformal field theories we have seen that proving the non-existence of a state-operator map is challenging. It is worthwhile to restrict ourselves to holographic theories to see if some mileage can be gained in this more constrained setup. By holographic CFT, we will mean any large $N$ CFT whose dual has Einstein gravity as its low energy effective theory. It turns out that the question of whether compact path integrals can prepare the vacuum state is related to a conjecture by Horowitz and Myers [13]. Their conjecture goes as follows:

Consider the $d + 1$ dimensional asymptotically AdS metric,

$$ds^2 = -\frac{r^2}{\ell_{AdS}^2}dt^2 + \frac{\ell_{AdS}^2 dr^2}{r^2\left(1 - (L_1/r)^d\right)} + r^2\left(1 - (L_1/r)^d\right)d\phi_1^2 + \frac{r^2}{\ell_{AdS}^2}d\phi_j^2, \tag{4.39}$$

which has a toriodal boundary with cycle lengths $L_i$ that are ordered as $L_1 < L_2 < \cdots < L_{d-1}$. It is a solution to the Einstein equations with negative cosmological constant in which the $L_1$ cycle is contractible in the bulk. Horowitz and Myers conjectured that this metric is the global minimum of the energy functional on the space of solutions with toroidal boundary conditions. In the original paper, Horowitz and Myers show that this solution sits at least at a local minimum of the energy functional and that the solution is stable to metric perturbations. Attempts to prove that it is a global minimum have failed thus far even though the conjecture continues passing more and more checks [16, 17]. One of the biggest difficulty that one faces when trying to prove the conjecture is that the techniques used to prove the positive energy theorem for asymptotically flat spaces and spacetimes with negative cosmological constant do not carry over. In those cases, a covariantly constant spinor could be defined at infinity, which is essential in the proof [18]. For a torus at infinity, such a spinor cannot be defined because of the boundary conditions we chose. Spinors need to be antiperiodic when going around the non-contractible cycles and therefore cannot be covariantly constant.

Some progress has been made since the original conjecture. For example, one can prove the uniqueness of the AdS soliton [17]. Suppose one is given a metric $g$ that satisfies the static vacuum Einstein equations, has pointwise negative mass and satisfies some condition near the conformal boundary as specified in [17]. If $g$ has the same cycle lengths for the boundary torus as in (4.39) and the same circle contracts in the bulk, then $g$ is isometric to the metric in (4.39). This is not quite a proof of the Horowitz-Myers conjecture, because it could very well be that there is another filling of the bulk that has even lower energy than the AdS soliton, but it comes close.

It is interesting to note that the Horowitz-Myers conjecture is essentially the gravitational dual of the CFT statements we are trying to prove, specialized to holographic CFTs. If one can prove that the AdS soliton is the global minimum of the energy functional, then it proves that for a holographic CFT, there is no compact path-integral that can produce the ground state because the soliton corresponds to an intrinsically non-compact Euclidean manifold. The argument can also be turned the other way around: proving that no compact manifold can produce the vacuum in CFTs would essentially prove the Horowitz-Myers conjecture from the field theory, by simply applying the theorem to holographic CFTs.

It is interesting to see that our conjecture is quite directly connected to an old conjecture in General Relativity. It is perhaps surprising to see that such a theorem has been notoriously hard to prove. It would be very interesting to see whether some CFT techniques we developed here could be used to gain mileage on the GR proof, or alternatively what the more recent progress on the GR side can teach us about CFT correlators of the stress tensor. We hope to return to these questions in the future.

# 5 Conclusion

In this paper, we have discussed the possibility of building a state-operator correspondence for CFT states living on a spatial two-torus. We took as a starting point that we needed to find a compact Euclidean manifold preparing the ground state for such a state-operator correspondence to work. Excited states would be built by adding operator insertions to this path integral, and we argued that torus states would in fact be more naturally associated to line operators.

We then investigated whether it is actually possible to build the ground state by a compact Euclidean path integral. We studied a particular example where the state is built by the path integral over a hemisphere times a circle. We showed that the geometry doesn't prepare the ground state by studying two examples: a free scalar theory and a holographic CFT.

The remainder of the paper was dedicated to answer the question: in a generic CFT, can a compact manifold with boundary two-torus prepare the ground state of the theory? We showed evidence that it could not. The vacuum should at least be a local minimum of the energy functional under deformations due to the insertion of operators. This means that certain integrated two-points function must vanish. We tied this fact to the existence of a conformal Killing vector with normal boundary conditions on the surface where the state is prepared. We showed that only a non-compact manifold with topology $\mathbf{R}_+ \times \mathbf{T}^2$ could have such a conformal Killing vector. We were however not able to formally prove that vanishing first order deformations implied the existence of a conformal Killing vector. Still, several arguments lead us to believe that it is the case and we leave this statement as a conjecture.

As a special case, one can consider holographic CFTs. This lead us to make connections with an old conjecture by Horowitz and Myers stating that the AdS Soliton is the geometry with minimal energy for a toroidal boundary. Our CFT arguments essentially provide a CFT version of the Horowitz-Myers conjecture, although they can be applied to more general CFTs.

It would be very interesting to extend our heuristic arguments to a full mathematical proof, which we have unfortunately not been able to do. Perhaps our argument could give new insights into the gravity attempts to prove the Horowitz-Myers conjecture. This could then provide a proof for holographic CFTs. It is also interesting to note that both field theory and gravity attempts at a proof seem difficult. Perhaps there is a particular reason for this, since holography usually evades the conservation of misery principle where hard problems on one side can be easier on the other.

It is also important to mention that we made the rather natural assumption that the vacuum should be prepared by a compact Euclidean path integral without the insertion of any operator. This is certainly how the state-operator correspondence works for states on the sphere and seems like a natural expectation to have for the torus as well. Nevertheless, it remains an assumption. It is extremely difficult to test the validity of this assumption. As a first step, one could consider the hemisphere state and insert local operators inserted at the south pole, but smeared over the non contractible circle direction. Such states could in principle have a lower energy expectation value than the empty hemisphere geometry. To test whether such states extremize the energy functional, one should then add further operators to the path integral and check whether the variations vanish. This would correspond to studying thermal four-point functions (one stress-tensor, two smeared operators $\mathcal{O}$ and a probe operator $\mathcal{O}'$ for the variation). Demanding that all such correlation functions vanish again seems too much to ask, but it is hard to rigorously prove. It is very interesting to note that extremizing the energy functional seems to be related to many positivity constraints on Euclidean correlation functions, all of which cannot be independent. It would be interesting to understand this connection better, but we leave this for future work.

In this paper, we have treated CFTs on general grounds and have not paid any particular

attention to subtleties that arise when we consider fermions. It is worth noting that if the theory does contains fermions, the statement can be rigorously proven quite easily in a particular sector that we didn't discuss. If instead we impose periodic boundary conditions for the fermions around both cycle of the two-torus, then it is obvious that we cannot build the vacuum with a compact Euclidean manifold with no insertions where one of the cycles shrink since it would violate the boundary conditions for the fermions[7]. In the case of CFTs with fermions, the results in this paper should therefore be seen as relevant when we impose anti-periodic boundary conditions around both cycles. The same argument in fact holds whenever there is a non-trivial boundary condition along any non-contractible cycle in the spatial manifold $\partial M$. For example, if we consider twisted sectors in orbifold theories [9], it is clear these can never be obtained via a path integral over a smooth Euclidean cap in which the circle is contractible, even if the cap has an arbitrary number of local operator insertions.

It would also be very interesting to study line operators directly, for example to understand what happens when they come close to one another or to local operators. For the usual state operator correspondence with states on the sphere, multiple operator insertions in the Euclidean cap can always be mapped to a linear combination of operators inserted at the origin and hence to a linear combination of the known states. It would be interesting to understand how similar ideas apply to line operators [19]. For free field theory, it would be interesting to try to explicitly do the relevant path integrals and determine the boundary condition near the line operator which would correspond to the ground state wave-functional on the boundary two-torus.

One could also explore a weaker version of the state-operator correspondence and investigate whether there is a construction that produces a dense set of states in the CFT Hilbert space using a compact path integral with operator insertions. From an axiomatic quantum field theory perspective, this seems very closely connected to the Reeh-Schlieder theorem [20]. Even though it has not been proven in curved space, there is a strong believe that such a theorem should exist [21](see also [22]). The Reeh-Schlieder theorem involves the insertion of operators on a spatial slice of a Lorentzian manifold. This is slightly different than demanding what states can be produced by a compact Euclidean path integral with operator insertions localized on some small region. For states on the sphere, it is straightforward to use the usual state-operator correspondence to see that one can produce a dense set of states in the Hilbert space by considering operator insertions near the south pole. It would be interesting to understand whether this is true on the torus. No argument we have given seems to indicate the contrary, and our intuition pushes us to think one could produce states arbitrarily close to the vacuum by inserting enough stress tensors near the south pole of the hemisphere state. The effect of these stress tensors would be to effectively create a long Euclidean throat, which is making the state closer and closer to the vacuum. It would be very interesting to see if one could make this idea precise.

A more general question is to consider Euclidean manifolds with two boundaries, $\partial M = \Sigma_1 \cup \Sigma_2$. The path integral over $M$ will then yield a map from $\mathcal{H}_{\Sigma_1}$ to $\mathcal{H}_{\Sigma_2}$. Do these maps have kernels or cokernels, and if so, are these generic or accidental or sectors labeled by particular quantum numbers?

Taking a step back, we can discuss the implications of the non-existence of a state-operator correspondence, should one be able to produce a proof. At first sight, it would seem to imply that there is no direct connection between modular invariance of the torus partition function and any statement about operators on the plane. Also, it seems to imply that there is no obvious way to get CFT torus correlators from the data of all operators (local or not) on the plane. It would be rather peculiar, especially for Lagrangian theories, if the theory on the plane would not fully specify the theory on the torus, and it would be interesting to investigate this further.

---

[7]We thank Rob Myers for discussions on this point.

Finally, there are known quantum field theories that do have a state operator correspondence on the torus. These theories are topological, like for example Chern-Simons theory [23][8]. These theories have no propagating degrees of freedom and are at first glance very different from generic CFTs. Nevertheless, it would be very interesting to understand better why the state-operator correspondence on the torus works there and not for CFTs from general principles. For example, it would be interesting to understand the role that local degrees of freedom play in this difference between CFTs and topological theories.

# Acknowledgement

We would like to thank Daniel Friedan, Rob Myers, Harvey Real and Edgar Shaghoulian for discussions. AB is supported by the NWO VENI grant 680-47-464 / 4114. JK is supported by the Delta ITP consortium, a program of the Netherlands Organisation for Scientific Research (NWO) that is funded by the Dutch Ministry of Education, Culture and Science (OCW).

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
