# Peer review of "Comments on a state-operator correspondence for the torus"

_SciPost Physics, doi:SciPost Phys. 5, 060 (2018)_

## Round 1 · Referee Report · Anonymous · 2018-5-2

Strengths
1) Original analysis on a not so explored important subject
Weaknesses
1) Methodology not rigorous
2) Inconclusive results
Report
Great progress has been achieved in the last years in the study of CFTs using the conformal bootstrap.
This method uses crossing symmetry to constrain the local data of a CFT, the spectrum of local primary operators and their OPE coefficients.
However, other constraints are expected to arise by demanding consistency of the CFT when defined on other manifolds.
A notable example of this sort is modular invariance in 2d CFTs on tori. In contrast to the two-dimensional case, not much is known about the nature
of such constraints for CFTs in $d>2$.
Motivated by these considerations, the authors investigate the existence of a state-operator correspondence in CFTs defined on manifolds of the form
$T^{d-1}\times R$, generalizing the known correspondence on the cylinder $S^{d-1}\times R$. The authors did not reach a definite answer, but provide
some evidence for a negative one, through examples and heuristic arguments. They also point out a connection between their arguments and a conjecture
by Horowitz and Myers, ref.[12].
The paper analyzes important and not so much explored ``global" aspects of CFTs. The paper is well written, though it is sometimes a bit sloppy.
The final results are inconclusive, but the arguments provided might be interesting enough for publication in Scipost.
Before publication, however, the authors should improve their presentation in two aspects:
1)- The implications of the alleged non-existence of the state-operator correspondence for tori to the possible global constraints in CFTs --- the starting motivation of the paper --- should be better and more extensively explained.
2)- The explanation of the connection of the paper with the Horowitz-Myers conjecture, limited to a few lines at the end of page 23, should be expanded and improved.
Requested changes
See report
Jorrit Kruthoff on 2018-04-07 [id 237]
Please note the following typo in the text: In the second line below eq. 3.10 we say 'In what follows, it will be convenient to make the smaller circle contractible...', but this should be 'the larger circle contractible...'.

---

## Round 1 · Referee Report · Anonymous · 2018-6-5

Strengths
1- The author investigate an interesting question, addressing a very relevant and modern subject.
2- The approach is original and can lead to further progresses in the field.
3- The discussion is organised in increasing complexity, first explaining a simple example and then proceeding with the general argument.
Weaknesses
1- The rigour of the discussion is a bit low and not up to the subject.
2- The authors could not produce a general proof of their argument and leave the main point of the paper as a conjecture.
Report
The manuscript investigates the existence of a states-operators correspondence for Conformal Field Theories defined on a more general manifold than the flat one, namely $S_d\times \mathds R$. In particular they focus on manifolds whose spacial slices are two dimensional tori.
As preliminary step, the author explore the possibility to create the vacuum state. They assume that the vacuum state is created by a Path Integral over a compact manifold with no insertion of operators. Despite the assumption is natural, it is not justified.
Next, an example is considered: the half two-sphere times a circle. In this case the authors manage to obtain (under a few assumptions) a relation between the thermal energy and the Casimir energy of the ground state. They then show in two realizations, the free boson and holographic CFTs that this two quantities differ. They conclude that this particular manifold doesn't admit a states-operators correspondence.
Despite the examples are quite instructive, they lack of rigour. In particular the authors do not show that all the assumptions made in the previous section are fulfilled.
The subsequent part of the paper is an attempt to disprove the existence of the state operator correspondence for a generic manifold with boundary $T_2$. The authors manage to prove that if a manifold satisfies certain properties (existence of a conformal killing vector normal to the torus) then it can create the vacuum state. They also point out that that the only manifold with this properties is non-compact and therefore is not a good candidate according to their original assumptions.
The other direction of the theorem is instead left as a conjecture. This is unfortunate since it is the most important part.
In conclusion, the work represents an original attempt to solve a very important and actual issue. It clarifies the problem and reduces it to a geometrical one.
The methods presented can spur further developments in the field.
After a minor revision, I think the work will qualify for a publication on SciPost.
Requested changes
Before publication I would like to see addressed the following points:
1- Comment on the assumptions of compactness of the manifold and insertion of operators in the Path Integral: why these are valid assumptions to create a the vacuum state and what happens if one relaxes them.
2- Show explicitly that the two examples presented satisfy all the assumptions made in Section 3 and comment on why these examples do not work.
3- Find a minimal set of assumption that allow to prove the second part of the theorem in Section 4.

---

## Round 2 · Referee Report · Anonymous · 2018-10-30

Report

The authors have improved their presentation as requested.
I think that the revised manuscript can be published in Scipost.

---

## Round 2 · Author Response

We thank the referees for the interesting comments. We agree with the referees that not all aspects of our discussion are completely rigorous, which is precisely why we were not able to produce a proof (or theorem) that a state-operator correspondence cannot exist for the torus. Formulating actual theorems in quantum field theory is notoriously difficult so it is perhaps not so surprising. Nevertheless, we believe we have made completely explicit what we were able to prove and what only constitutes evidence towards a possible proof.
We have added two paragraphs before section 2.1 to further discuss why we consider compact Euclidean manifolds as well as a discussion of the assumption of considering manifolds without operator insertions. It is true that we were not able to prove that such configurations are even local minima of the energy functional. In fact, we believe that they are not global minima of the energy functional and they cannot prepare the vacuum on the torus. The vacuum is most likely only prepared by the half infinite line times the torus, but one can presumably get very close to the vacuum by considering a hemisphere with many stress-tensors inserted near the south pole, effectively creating a long Euclidean throat.
In connection to this, we added a paragraph in the discussion connecting our approach to the Reeh-Schlieder theorem, as an attempt to see how one can compare the approach in this paper to one of the few proven theorems in field theory. It is interesting to note that even the Reeh-Schlieder theorem has not been proved in curved space.

---

## Round 2 · List of Changes

We have modified the specific points addressed by the referees in the following way:
1. Point 1 of referee 1: we added a discussion of the implication of the absence of a state-operator correspondence in the discussion section on page 27.
2. Point 2 of referee 1: we extended the section on the Horowitz-Myers conjecture.
3. Point 1 of referee 2: we added two paragraphs above section 2.1 on page 5 discussing these assumptions. We would like to emphasize again that a compact manifold is a necessary ingredient for a state-operator correspondence. The boundary condition at t = −∞ must be replaced by a lower codimension boundary condition. On the other hand, the assumption about not adding any operators to the path integral was a starting assumption. We do not believe that such configurations are global (or even local) minima of the energy functional.
4. Point 2 of referee 2: we did not make any assumptions in this section, we simply computed the expectation value of the energy in the hemisphere state for two theories: the free boson and a holographic CFT. In both cases, we found that the energy is always greater than the known vacuum energy on the two-torus. This shows that the hemisphere state is not the right candidate to produce the vacuum in all CFTs.
5. Point 3 of referee 2: this is a hard question. We failed to provide a necessary set of conditions such that there does not exist a state-operator correspondence. Providing sufficient conditions is much easier, and we gave such an example in section 4.4. If we assume sufficiently nice analytic properties of the correlation functions, then one naturally obtains a conformal killing vector on the Euclidean section. Nice analytic properties of the Lorentzian correlator thus provides a sufficient condition to prove the other direction of the theorem. However, we would like to emphasize that we do not know whether it is natural to impose such a nice analytic property of the correlator, since their analytic properties can be quite subtle in quantum field theory. It would be very interesting to try to understand this question in more detail.

---

## Editorial Decision

published